# Laser Tooth Preparation for Pit and Fissure Sealing

**DOI:** 10.3390/ijerph17217813

**Published:** 2020-10-26

**Authors:** Yair Schwimmer, Nurit Beyth, Diana Ram, Eitan Mijiritsky, Esti Davidovich

**Affiliations:** 1Department of Pediatric Dentistry, Hebrew University-Hadassah School of Dental Medicine, 91120 Jerusalem, Israel; yschwimmer@yahoo.com (Y.S.); dianar@ekmd.huji.ac.il (D.R.); 2Department of Prosthodontics, Hebrew University-Hadassah School of Dental Medicine, 91120 Jerusalem, Israel; nuritb@ekmd.huji.ac.il; 3Department of Otolaryngology Head and Neck Surgery and Maxillofacial Surgery Tel-Aviv Sourasky Medical Center, Sackler School of Medicine, Tel Aviv University, 6139001 Tel-Aviv, Israel; mijiritsky@bezeqint.net

**Keywords:** pits, fissures, sealants, lasers

## Abstract

*Objectives:* Various approaches are available for pit and fissure sealing, including: the use of sealants, with or without mechanical preparation; the use of etching, with or without bonding; and the use of lasers as an alternative to mechanical preparation. The objective of this study is to evaluate pit and fissure sealing by comparing the retention and microleakage of sealants, between mechanical and Er:Yag laser enamel preparation. *Methods:* Sixty extracted sound third molars are classified into six groups: A, bur mechanical preparation and sealant application; B, bur mechanical preparation, etching and sealant; C, bur mechanical preparation, etching, bonding and sealant; D, laser mechanical preparation and sealant; E, laser mechanical preparation, etching and sealant application; F, laser mechanical preparation, etching, bonding, and sealant. Statistical analysis methods include Fisher’s exact test, a general linear model for one-way analysis of variance (ANOVA) of multiple comparisons, and Bonferroni multiple comparison tests. *Results:* All the groups showed dye microleakage beneath the sealants. Less microleakage was observed for those that used bur rather than laser, 41 versus 44 specimens, respectively. The number of specimens without microleakage decreased as follows: group E (24), group A (18), groups B and F (17), group C (14), and group D (5). Retention was 100% in all groups except group D. Conclusion: Mechanical preparation increases retention of sealants, especially when etching material is used; additionally, bonding can help the retention. The best technique is mechanical preparation via laser and subsequent use of etching, without bonding prior to application of the dental sealant.

## 1. Introduction

Dental pit and fissure sealing are considered effective noninvasive approaches to preventing caries in children and adolescents. Differences in the caries prevalence of the populations examined may account for differences in the effectiveness of these measures. According to a Cochrane review published in 2013, sealing of the occlusal surfaces in permanent molars in children and adolescents reduces caries for up to 48 months in comparison to not sealing [1]. Further, sealants have been recommended for people with several caries, high caries risk assessment, deep narrow pits and fissures of deciduous and permanent molars and premolars, and non-cavitated lesions [2,3,4,5,6]. Beauchamp et al. reported that sealants can reduce caries by 59–96% during follow-up periods of 1–9 years [5]. Studies with longer follow-up showed a reduction in the quantity and quality of the evidence obtained [1].

Over 85% of the lesions in the permanent dentition involve surfaces with pits and fissures, despite the availability of preventive measures such as sealants [7]. Molars and premolars are the teeth most susceptible to caries development due to their occlusal surface morphology, which limits the amount of saliva and makes it difficult to clean with a toothbrush. Pits and fissures are therefore the areas that are most prone to caries and need particular protection to prevent carious lesions [2].

Sealants are composed of various materials and are inserted using a number of different techniques. Resin, one of the most common materials used for dental sealant, contributes to preserving the integrity of the occlusal surface and acts as an effective mechanical obstacle for plaque retention, which, in turn, reduces the number of incidences of fissure caries [8,9,10]. The efficacy of sealing procedures depends on using the correct application technique. A number of operative protocols have been suggested in the literature that prolong protection against caries [2], for example, sealant can be inserted without any mechanical preparation, only using etching (with or without bonding). Alternatively, sealant can be inserted using mechanical preparation via air abrasion and other protocols. For sealants placed on maxillary molar teeth, the retention rate was higher when the insertion involved mechanical preparation [11].

Microleakage is examined by penetration of dye that enters between the tooth surface and the sealant material. Entrance of the dye may illustrate the penetration of the cariogenic bacteria beneath the sealant, which causes dental caries. The sealant may fracture or fall due to the microleakage [1]. 

Lasers can be used as an alternative for mechanical preparation (enameloplasty) before sealing pits and fissures. One study reported that adolescents preferred the Erbium YAG laser for carious tissue removal and perceived it as more comfortable than conventional mechanical preparation [12]. Evidence shows a higher sealing outcome when bur preparation, acid etching, or air abrasion is performed prior to application of the sealant [13]. Adding a bonding agent layer between the sealant and the enamel that has been contaminated with saliva was shown to increase bond strength and retention of the resin sealants and possibly improve the success of all sealant applications [14].

The relative effectiveness of the available approaches for enamel sealing has yet to be established. The aim of the present study was to assess retention and microleakage of pit and fissure sealants following various methods of preparation and application of materials. We hypothesized that less retention and more microleakage would be attained following a protocol based on laser enamel preparation, without the use of etching or adhesive material prior to sealant placement.

## 2. Materials and Methods

The protocol of the study was approved by the Institutional Review Board for Human Clinical Trials of Hadassah School of Dental Medicine (Identifier: 0483-15-HMO). 

During January–May 2018, 60 sound extracted third molars, which were acquired from the Department of Oral Maxillofacial Surgery, were stored in saline solution, then cleaned with water and a soft brush (low speed air motor hand piece). Molars without any decay or fractures were chosen. The apices were sealed with self-cured glass ionomer. The teeth were arbitrarily assigned to six equal groups of ten specimens each, following the preparation protocol presented in Figure 1. The surface of the enamel was mechanically prepared by 330 high speed bur (SS WHITE, Lakewood, NJ, USA) touching the surface, without implementing any dental cavity preparation. The teeth were mechanically prepared by the Er:YAG laser by directing the light of the laser to the enamel surface until it appeared white, without making a cavity (Figure 1).

Etching (Super-Etch™ 37% phosphoric acid etch, SDI^TM^, Bayswater, Victoria, Australia) was performed for 20 s, followed by 10 s washing and 5 s drying using a triple syringe. Two layers of bonding (Single bond universal™, 3M ESPE, St Paul, MN, USA) were applied and followed by 5 s of air application using a triple syringe, following manufacturer instructions. The sealant (Clinpro™ Sealant, 3M™ ESPE™, St Paul, MN, USA) application was applied by homogenous spreading of the material using a dycalon and then polymerized for 20 s by D-Light Pro Dual wavelength high-power LED curing light (1400 mW/cm^2^) (GC Europe™, Leuven, Belgium). All the materials and instruments used in the present study are listed in Table 1.

Retention of the sealants was tested with a dental explorer by four specialists in pediatric dentistry who were blinded to the groups of the teeth that were examined. The results were recorded either as total retention, partial loss, or total loss. Sealant failure was defined when the material was partially or totally lost, as previously described [11]. The experiments required examination with dental probing; however, this did not loosen the sealant.

To assess inter-examiner reliability (kappa = 0.95 between examiners), each examiner evaluated retention of the same 10 teeth prior to conducting the experiment.

The teeth were subjected to 500 thermo-cycles in water baths with a temperature in the range of 7–55 °C (the duration of each bath was 30 s and the time between baths was 5 s). The teeth were dried, then sealed entirely with three layers of nail varnish, separate from the sealant and 1 mm beyond the sealant margins. All the teeth were immersed in 1% methylene blue solution for 24 h to enable penetration of the dye into possible gaps between the tooth substances and the sealant. After the teeth were removed from the dye, the nail varnish was removed using a sharp instrument. The specimens were dried and embedded in self-cured methyl methacrylate resin (UNIFAST Trad, GC™ America Inc. Alsip, IL, USA). The teeth were each measured and divided equally into three longitudinal bucco-lingual sections with a water-cooled electric diamond saw. Retention and microleakage were evaluated: (1) clinically, using a dental explorer; (2) using a stereo microscope (NIKON SMZ25) with a magnification of 25× for examination of the teeth, and microleakage penetration depth was recorded in micro-millimeters; and (3) by computer measurements of the photographic depiction of the dye penetration.

The penetration of the dye was evaluated in the enamel and sealant interface. The length of the sealant and the length of the dye penetration were recorded and the percent of dye penetration of the total length of the sealant was calculated using the ruler of the microscope software.

### Statistical Analysis

All statistical analyses were performed using SAS^®^ software (SAS Enterprise Guide 7.1). Descriptive statistics were performed to assess the differences between the preparation methods. Fisher’s exact test was used to evaluate the presence or absence of microleakage (categorical parameter). A general linear model was used for one-way analysis of variance (ANOVA) of multiple comparisons of the means of continuous parameters (length of the sealant, length of dye penetration, and percent of dye penetration) between preparation methods. Bonferroni multiple comparison tests were used to determine statistically significant differences between all the methods. The level of significance for all tests was set to 0.05.

## 3. Results

In all groups except for group D, retention of the sealants was scored as 100%. Group D was characterized by mechanical preparation with Er:YAG laser and sealant application. Nine of the teeth in this group were scored as total retention and one tooth was scored as partial sealant retention. Between the four examiners, all the evaluations were the same, yielding 100% inter-examiner reliability.

Microscope depiction demonstrated dye penetration in all six groups. From 60 teeth, 180 specimens were prepared. Of them, 95 had no penetration at all and 85 had partial penetration, including some with total penetration of the dye beneath the sealants. No statistically significant differences were observed between groups A, B, C, E, and F. However, a statistically significant difference was found in dye penetration between the preparation methods; the highest dye penetration was found in group D compared to the other five groups (*p*-value < 0.0001).

Retention of the sealants was not found to differ significantly between the groups with high speed bur preparation (groups A–C) and those with Er:YAG laser preparation (groups C–E) (*p*-value = 0.20). However, when high speed mechanical preparation was used, dye penetration was less beneath the sealant: 46 versus 49 teeth. The order of penetration of the dye from the highest to the lowest group was as follows: D > C > F/B > A > E (Table 2).

Dye penetrations are shown for the six groups examined (A–F). See Figure 1 for details of these groups.

For groups A and B, the penetration of the dye was 30% of the length of the sealant; in group C, 20%; and in groups E and F, 10%. The worst dye penetration was in group D, with a mean of 70%. (Figure 2)

The Bonferroni multiple comparison test indicated statistically significant differences in dye penetration between the preparation methods (*p*-value < 0.0001). Treatment D demonstrated the highest dye penetration. No significant differences were detected between the other preparation methods (*p*-value > 0.05), as presented in Table 3.

Figure 3 shows a specimen from each group, with the dye leakage beneath the dental sealant. The lengths of the sealants were 1019.98, 1639.55, 3344.3, 1859.74, 2502.92, and 2149.77 µm for specimens A–F, respectively. The proportions of the teeth that were penetrated were 100% in groups A, B, and D; 39.3% in group C; 78.2% in group E; and 54.2% in group F.

## 4. Discussion

Bacteria colonization in morphologically susceptible areas such as young molar teeth can exacerbate demineralization. Since bacteria can take advantage of pits and fissures to colonize enamel, preventing infection is extremely important [15]. The American Academy of Pediatric Dentistry and the American Dental Association concluded that dental sealants are underused by practitioners in the prevention and treatment of early stages of dental caries [16]. Accordingly, they published guidelines that strongly recommend the application of sealants on the occlusal surfaces of primary and permanent teeth. Resin-based sealant showed better retention than other materials. The exception to this was the application of glass-ionomer sealants for erupting teeth, thus serving as temporary sealant until the full exposure of teeth. The guidelines state that sealants should be placed according to the instructions of the manufacturer of the material used. This includes the recommendation of etching the surface before application of the resin-based material and the use of enameloplasty or adhesive materials [16].

Enameloplasty and adhesive materials applied prior to the sealant increase the retention of the sealant and decrease the microleakage between the sealant and the pit. However, some reports concluded that mechanical preparation was not beneficial [16,17], as evidenced by increased incidences of caries in teeth sealed after mechanical preparation with these materials. Loss of sealants seems to predispose teeth to the development of caries, especially when teeth preparation is performed. With regard to primer placement before sealant application, one randomized clinical trial found that acetone or ethanol solvent-based primers, especially the single bottle system, enhanced the retention of sealants. In contrast, water-based primers drastically reduced the retention of sealants [18]. With regard to self-etch bonding agents that do not involve a separate step for etching, a systematic review concluded that self-etch bonding agents may provide a lower quality of retention than the acid etch technique [19]. However, a randomized clinical trial reported similar retention rates of self-etching and acid etching [20]. Hence, adhering to the manufacturer’s instructions for each sealant material is important [21]. Notably, due to the high discrepancy in results (in mm), which reflects differences in teeth size and in fissure sealants, the results are presented as the difference between the groups in the percentage of dye penetration. Interestingly, we found lower retention of sealant following mechanical preparation with the Er:YAG laser without etching and bonding (90%) compared to other protocols of mechanical preparation. The differences between the groups were not statistically significant, except for the difference between the group with laser mechanical preparation and sealant and the group with the highest microleakage. This supports our hypothesis of less retention and more microleakage following a protocol that did not include etching or adhesive material prior to sealant placement. Retention of the dental sealants that were placed on sound teeth was high in all groups, and our findings regarding sealant retention agree with previous in-vitro and in-vivo studies [22,23].

A possible explanation for our findings relates to the lesser penetration of the sealant into the enamel prisms when using phosphoric acid. From this, we can assume that without the use of etching material, the sealant material would harden better after enameloplasty with bur than with the Er:YAG laser [24].

We observed more microleakage following the use of the Er:YAG laser in comparison to bur (44 vs. 41). This concurs with the research of Borsatto et al. and Yossef et al. [8,25]. In both these studies, as in our study, the protocol of tooth surface preparation consisting of Er:YAG laser, without etching and bonding, was the only protocol that did not show full retention. We suggest that that this may be due to insufficient ablation by the laser of the tooth surface and the lack of extension of the enamel prisms. Therefore, we do not recommend using this protocol, and advocate the use of etching, with the goal of increasing retention of the sealant.

Of note, Topaloglu-Akand et al. [26] reported leakage with all the protocols they examined. The researchers examined 96 molar teeth in which enamel was prepared with Er:YAG laser and different materials were applied: 1. Er:YAG laser; 2. Er:YAG laser + 37% H_3_PO_4_ (15 s); 3. ER:YAG laser + 37% H_3_PO_4_ + Prime and Bond NT; 4. Er:YAG laser + G Bond; 5. Er:YAG laser + Prime and Bond NT; 6. 37% H_3_PO_4_; 7. 37% H_3_PO_4_ + Prime and Bond NT; 8. G Bond. Sealant material (Clinpro, 3M ESPE, Seefeld, Germany) was applied. They found that the Er:YAG laser showed the highest microleakage scores, whereas the Er YAG laser + 37% H_3_PO_4_ showed the lowest. Although the 37% H_3_PO_4_ group showed higher scores than the Er:YAG laser + 37% H_3_PO_4_, the difference was not statistically significant. The authors concluded that only etching without enameloplasty is sufficient. In contrast, Javadinejad et al. [27] found that mechanical preparation with pre-sealant application does not improve the mechanism of the adhesive. Nevertheless, they upheld their support for the use of etching. Memarpour et al. showed less microleakage in the group treated with the Er:YAG laser than following etching, bonding, and sealant application. However, they did not find a statistically significant difference between bur mechanical preparation with and without bonding. Thus, they concluded that the Er:YAG laser can be an alternative method before sealant application [28].

Our findings contrast with those of Nahvi et al. [29]. They reported significant reduction in microleakage with the use of a pre-sealant bonding agent. Elsewhere, the bonding agent beneath the sealant was shown to increase the bond strength even in moisture-contaminated conditions, and to decrease microleakage [30]. This is similar to our findings, though our results did not show statistical significance. Notably, Tehrani et al. recommended the use of a combination of etching and bonding agents prior to sealant placement, as the best technique for sealing pits and fissures [31].

There are some limitations to this research. The inclusion of only 10 teeth in each group makes achieving statistical significance difficult. Additional studies should be performed with larger groups of teeth and using different materials to further elucidate the best method of tooth preparation for sealants.

In conclusion, we expect that the findings of this study may help clinicians select the preferable method for pits and fissure sealants. We concluded that mechanical preparation increases retention of sealants, especially when using etching materials as bonding can help the retention. The best technique was mechanical preparation via laser and the subsequent use of etching, without bonding prior to application of the dental sealant.

## 5. Conclusions

Our results showed high performance and no statistically significant differences between high speed mechanical preparation using laser and all other protocols examined. Nevertheless, the technique of laser mechanical preparation without etching and bonding is not recommended. 

## Figures and Tables

**Figure 1 ijerph-17-07813-f001:**
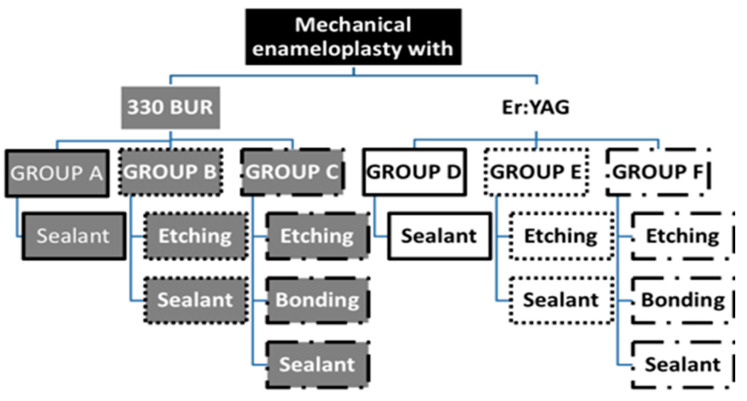
Group design. The preparations for the 6 groups (A–F). For groups A–C, 330 tungsten bur was used and Er:YAG laser for groups D–F.

**Figure 2 ijerph-17-07813-f002:**
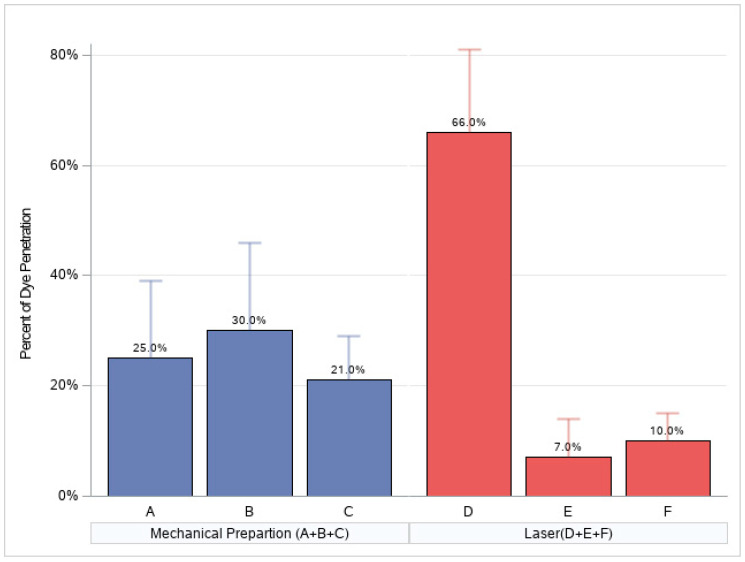
The percentage of the total length of the sealant that the dye penetrated following various protocols (the mean percentage is calculated as dye penetration).

**Figure 3 ijerph-17-07813-f003:**
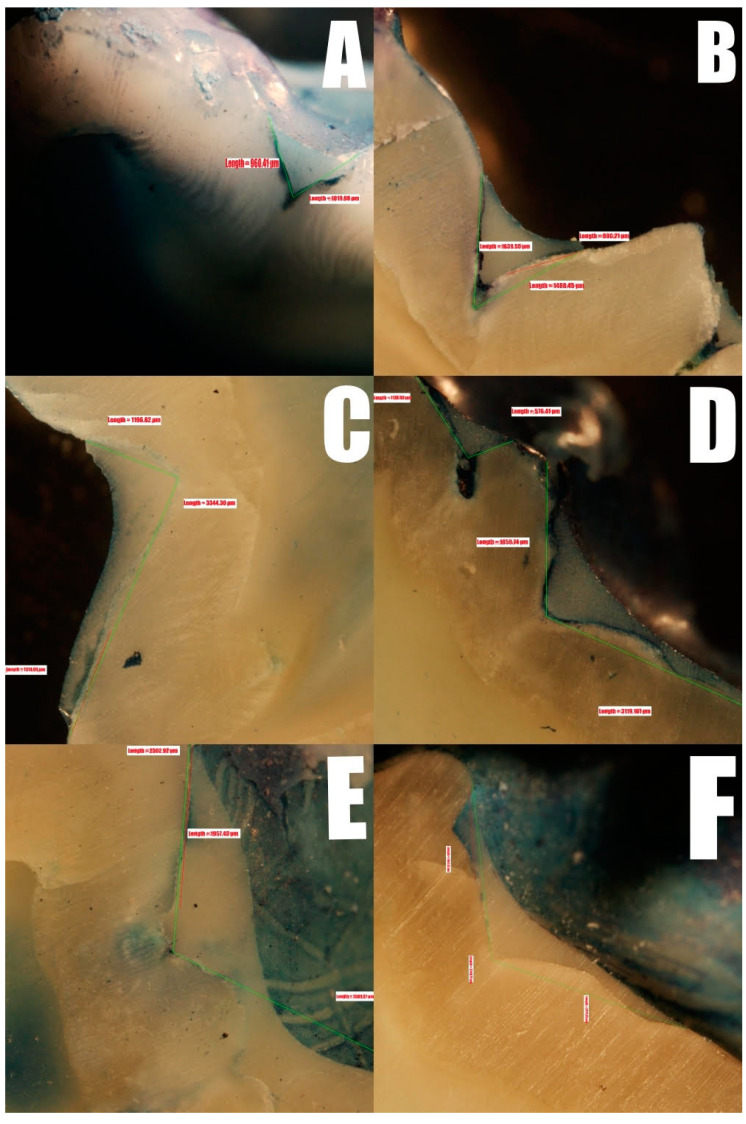
Representative depiction for each tested group.

**Table 1 ijerph-17-07813-t001:** Materials and instruments used in the present study.

Instruments/Materials	Company
Sealant	Clinpro™ Sealant (3M™ ESPE™, St Paul, MN, USA)
Etching	Super-Etch™ 37% phosphoric acid etch (SDI^TM^, Bayswater, Victoria, Australia)
Bonding	Single bond universal (3M ESPE, St Paul, MN, USA)
Er:YAG LASER	Syneron Dental Lasers (Opus 20 by Opus Dent, Netanya, Israel) with 7.5 w; 300 mJ; 25 Hz per pulse and 12 pulses per second
Thermocycling	ADA Health Foundation
Self-cured acryl resin	Unifast trad^®^ (GC Europe™, Leuven, Belgium)
Scanning stereo microscope	NIKON SMZ-25

**Table 2 ijerph-17-07813-t002:** Penetration of the dye beneath the dental sealant showing differences between the groups in the numbers of teeth with and without penetration. Penetration was compared between the groups using Fisher’s exact test.

Treatment								
		A	B	C	D	E	F	Total
**No Penetration**	Frequency	18	17	14	5	24	17	95
	Expected	15.8	15.8	15.8	15.8	15.8	15.8	
	Cell Chi-Square	0.3	0.09	0.21	7.4123	4.21	0.09	
	Percent	10	9.44	7.78	2.78	13.33	9.44	52.78
**Penetration**	Frequency	12	13	16	25	6	13	85
	Expected	14.2	14.2	14.2	14.2	14.2	14.2	
	Cell Chi-Square	0.33	0.1	0.24	8.28	4.71	0.1	
	Percent	6.67	7.22	8.89	13.89	3.33	7.22	47.22
	Total	30	30	30	30	30	30	180
	Percent	16.7	16.7	16.7	16.7	16.7	16.7	100

**Table 3 ijerph-17-07813-t003:** Mean values, standard errors (SEs), and Bonferroni multiple comparison tests for means of percentage of dye penetration between preparation methods.

Preparation Method	Mean ± SE	Comparison of Preparation Method	*p*-Value
A	25 ± 6.7	A vs. B	1.00
		A vs. C	1.00
		A vs. D	<0.0001 *
		A vs. E	0.4164
		A vs. F	0.9232
B	30 ± 7.7	B vs. A	1.00
		B vs. C	1.00
		B vs. D	<0.0002 *
		B vs. E	0.0727
		B vs. F	0.1915
C	20 ± 4.2	C vs. A	1.00
		C vs. B	1.00
		C vs. D	<0.0001 *
		C vs. E	1.00
		C vs. F	1.00
D	66 ± 7.3	D vs. A	<0.0001 *
		D vs. B	<0.0002 *
		D vs. E	<0.0001 *
		D vs. F	<0.0001 *
E	7 ± 3.3	E vs. A	0.9232
		E vs. B	0.1915
		E vs. C	1.00
		E vs. D	<0.0001 *
		E vs. F	1.00
F	10 ± 2.5	F vs. A	1.00
		F vs. B	1.00
		F vs. C	1.00
		F vs. D	<0.0001 *
		F vs. E	1.00

* Significant *p*-value; F-test results are presented; these indicate statistically significant differences in the percentage of dye penetration between the groups (*p*-value < 0.0001).Group A, bur mechanical preparation and sealant. Group B, bur mechanical preparation, etching and sealant. Group C, bur mechanical preparation, etching, bonding and sealant. Group D, laser mechanical preparation and sealant. Group E, laser mechanical preparation, etching and sealant. Group F, laser mechanical preparation, etching, bonding and sealant.

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
