# Peer review of "Laser Tooth Preparation for Pit and Fissure Sealing"

_ijerph, 2020, doi:10.3390/ijerph17217813_

Round 1

Reviewer 1 Report

Dear authors, 

Thank you for your revised version of your manuscript. Please find attached comments for your kind consideration. 

Sincerely, 

Reviewer 1

Author Response

Veronica You
Assistant Editor
International Journal of Environmental Research and Public Health

Re: Manuscript ID: ijerph-886953, entitled “Laser tooth preparation for pit and fissure sealing”.

October  2020

Dear Dr. You

We thank you again for the constructive responses and remarks of the referee, which have helped to further clarify our manuscript. Below are our point-by-point responses to their comments. In our revised manuscript, we marked all the changes with TRACK CHANGES.

 We hope that the present version of our manuscript will be found suitable for publication in The International Journal of Environmental Research and Public Health

We look forward to hearing from. 

Sincerely yours,

Esti Davidovich and co-authors

Reviewer #1:

  1. Comment 9: You have mentioned in your response that correction made in P3 Lines 89-90 (your old version pages and lines) but your correction with the IRB was made in the P3 L 83-84 (new version). It was confusing. Please be aware that for your future responses to reviewers, you need to include numbers of your altered pages and lines as well to avoid extra work for reviewers. Please add in the sentence full description "Institutional Review Board in P 3 L83-84

RESPONSE: The Institutional Review Board was added in P3 L83-85

  1. Comment 13 P3 L114-122: Instead of describing all steps in the methods, I wanted….

RESPONSE: Thank you we preferred to leave the detailed description as one of the other reviewers requested.

  1. Comment 18: Please be aware that for your future responses to reviewers, you need corrections in the P13 L 262-270. Need appropriate lines, where you made changes.

RESPONSE:  The protocol was added in detail, on page 11 L268-278

Veronica You
Assistant Editor
International Journal of Environmental Research and Public Health

Re: Manuscript ID: ijerph-886953, entitled “Laser tooth preparation for pit and fissure sealing”.

October  2020

Dear Dr. You

We thank you again for the constructive responses and remarks of the referee, which have helped to further clarify our manuscript. Below are our point-by-point responses to their comments. In our revised manuscript, we marked all the changes with TRACK CHANGES.

 We hope that the present version of our manuscript will be found suitable for publication in The International Journal of Environmental Research and Public Health

We look forward to hearing from. 

Sincerely yours,

Esti Davidovich and co-authors

Reviewer #1:

  1. Comment 9: You have mentioned in your response that correction made in P3 Lines 89-90 (your old version pages and lines) but your correction with the IRB was made in the P3 L 83-84 (new version). It was confusing. Please be aware that for your future responses to reviewers, you need to include numbers of your altered pages and lines as well to avoid extra work for reviewers. Please add in the sentence full description "Institutional Review Board in P 3 L83-84

RESPONSE: The Institutional Review Board was added in P3 L83-85

  1. Comment 13 P3 L114-122: Instead of describing all steps in the methods, I wanted….

RESPONSE: Thank you we preferred to leave the detailed description as one of the other reviewers requested.

  1. Comment 18: Please be aware that for your future responses to reviewers, you need corrections in the P13 L 262-270. Need appropriate lines, where you made changes.

RESPONSE:  The protocol was added in detail, on page 11 L268-278

Reviewer 2 Report

The data presentation has been improved. 

Please double check the Statistical analysis section. Especially that "no significant differences were detected between" E vs A, E vs. B and E vs. C appears odd. Based on this reviewer's knowledge, e.g. E vs B are significantly different . Of course this depends on which test one uses. This raises the question whether the test the authors used here is appropriate. Please look into it and justify.

Also, please reflect in the Discussion the causes of the high SE of percentage dye penetration, as was written in the cover letter. 

Author Response

Veronica You
Assistant Editor
International Journal of Environmental Research and Public Health

Re: Manuscript ID: ijerph-886953, entitled “Laser tooth preparation for pit and fissure sealing”.

October  2020

Dear Dr. You

We thank you again for the constructive responses and remarks of the referee, which have helped to further clarify our manuscript. Below are our point-by-point responses to their comments. In our revised manuscript, we marked all the changes with TRACK CHANGES.

 We hope that the present version of our manuscript will be found suitable for publication in The International Journal of Environmental Research and Public Health

We look forward to hearing from. 

Sincerely yours,

Esti Davidovich and co-authors

Reviewer #2:

  1. The data presentation has been improved. 

RESPONSE: Thank you very much

  1. Please double check the Statistical analysis section. Especially that "no significant differences were detected between" E vs A, E vs. B and E vs. C appears odd. Based on this reviewer's knowledge, e.g. E vs B are significantly different . Of course this depends on which test one uses. This raises the question whether the test the authors used here is appropriate. Please look into it and justify.

RESPONSE:

 Non -parametric, Wilcoxon rank sum test (Mann-Whitney U ,SASÒ PROC NPAR  1 way procedure ) test was applied to test the null hypothesis of no difference in  %Dye Penetration  between treatment groups  . Wilcoxon rank sum test was applied since the tested parameter, %Dye Penetration, is skewed and the   distribution do not follow normal distribution. Adjustments for multiple comparisons and multiple testing between treated group   were applied.

 The level of significance for all tests was set to 0.05.

 The adjustments of multiple comparison test indicate statistically significant differences in dye penetration between the preparation methods (p-value<0.0001).

Treatment D demonstrates the highest dye penetration. Treatment D demonstrates statistically significant different from all treatments groups.

Treatment

Wilcoxon Z

DSCF Value

Pr > DSCF

A vs. B

-0.2981

0.4216

0.9997

A vs. C

-0.2729

0.386

0.9998

A vs. D

-3.7572

5.3135

0.0024

A vs. E

1.9702

2.7862

0.3596

A vs. F

0.8753

1.2379

0.9525

B vs. C

-0.0319

0.0451

1

B vs. D

-3.4021

4.8114

0.0088

B vs. E

2.241

3.1693

0.219

B vs. F

1.0222

1.4456

0.9107

C vs. D

-4.1119

5.8151

0.0006

C vs. E

2.6531

3.752

0.0849

C vs. F

1.7037

2.4094

0.5293

D vs. E

5.4103

7.6514

<.0001

D vs. F

5.077

7.1799

<.0001

E vs. F

-1.6482

2.3309

0.5663

  1. Also, please reflect in the Discussion the causes of the high SE of percentage dye penetration, as was written in the cover letter. 

RESPONSE: The cause of high SE and percentage of dye penetration was added in the discussion P 10 L233-234

Reviewer 3 Report

Dear Authors,

thank you for you article which I found very interesting. I just have some minor revisions to propose to you in order to improve the quality of your word.

MATERIALS AND METHODS

Line 85: Please specify whether you considered specific inclusion and exclusion criteria when selecting the teeth for the study.

Line 90: Please specify the type of bur used (Manufacturer, city and country)

Line 99: please specify the lamp used for photopolymerization and the relative characteristics (intensity, Manufacturer, city and country)

Line 99: as regards Best-Etch™ 37% please add the city and country of the Manufacturer

Lines 97-103: I suggest describing the procedures with the following order: etching, bonding and sealant application.

DISCUSSION

I suggest starting your discussion with few general words on the importance of preventing bacterial colonization of the teeth. For example you could consider the following article:

Scribante A, Poggio C, Gallo S, Riva P, Cuocci A, Carbone M, Arciola CR, Colombo M. In Vitro Re-Hardening of Bleached Enamel Using Mineralizing Pastes: Toward Preventing Bacterial Colonization. Materials (Basel). 2020

Author Response

Veronica You
Assistant Editor
International Journal of Environmental Research and Public Health

Re: Manuscript ID: ijerph-886953, entitled “Laser tooth preparation for pit and fissure sealing”.

October  2020

Dear Dr. You

We thank you again for the constructive responses and remarks of the referee, which have helped to further clarify our manuscript. Below are our point-by-point responses to their comments. In our revised manuscript, we marked all the changes with TRACK CHANGES.

 We hope that the present version of our manuscript will be found suitable for publication in The International Journal of Environmental Research and Public Health

We look forward to hearing from. 

Sincerely yours,

Esti Davidovich and co-authors

Reviewer #3:

  1. Thank you for your article which I found very interesting.

RESPONSE: Thank you very much

  1. MATERIALS AND METHODS: Line 85: Please specify whether you considered specific inclusion and exclusion criteria when selecting the teeth for the study.

RESPONSE: The inclusion and exclusion criteria for selecting the teeth were added in P3 L88-89.

  1. Line 90: Please specify the type of bur used (Manufacturer, city and country)

RESPONSE: The type of the bur is now specified in P3 line 92.

  1. Line 99: please specify the lamp used for photopolymerization and the relative characteristics (intensity, Manufacturer, city and country)

RESPONSE: The lamp used for photopolymerization and the relative characteristics (intensity, Manufacturer, city and country) are now specified in P4 L105-106

  1. Line 99: as regards Best-Etch™ 37% please add the city and country of the Manufacturer

RESPONSE: The type of the etching is now specified in P4 lines 99-100

  1. Lines 97-103: I suggest describing the procedures with the following order: etching, bonding and sealant application.

RESPONSE: The order of the procedure was reorganized, accordingly. See lines 98-109.

  1. DISCUSSION: I suggest starting your discussion with few general words on the importance of preventing bacterial colonization of the teeth. For example, you could consider the following article:

Scribante A, Poggio C, Gallo S, Riva P, Cuocci A, Carbone M, Arciola CR, Colombo M. In Vitro Re-Hardening of Bleached Enamel Using Mineralizing Pastes: Toward Preventing Bacterial Colonization. Materials (Basel). 2020

RESPONSE: This reference was added and cited in the beginning of the discussion P 9 L 208-210 and P 13 L336-338.

This manuscript is a resubmission of an earlier submission. The following is a list of the peer review reports and author responses from that submission.

Round 1

Reviewer 1 Report

Dear authors, 

In overall, the manuscript is of importance for practicing dentists. Needed some improvements based on the comments given. 

With regards,

Author Response

Re: Manuscript ID: ijerph-886953, entitled “Laser tooth preparation for pit and fissure sealing”.

August 2020

Reviewer #1:
1) The content of the manuscript abstract has to be improved by enhancing quality of academic writing including in the methods and statistical analysis appropriately

 RESPONSE:

We revised the abstract, and improved the writing, including the methods and statistical analysis section.

2) Please add in the result of your abstract about outcome of your retention according to your objective

RESPONSE

We added the following sentence to the Results: “Retention was 100% in all the groups except group D.” 

3) In the abstract please connect your conclusion on retention with microleakage of sealants as well

RESPONSE:

The conclusion section in the abstract was re-written. P2 L33-36

4) Please check English grammar, logic and structure of sentences. For example 1, line 12-14, 29-30 P3 ( in the table : check the word "foundation"); P8 L198; P9 L 209 (comma or dot)?

RESPONSE

A professional English-speaking medical writer reviewed the entire manuscript and made corrections as appropriate, including the mistakes mentioned by the Reviewer.

5) please include in the introduction (P2) part information on microleakage from your literature review briefly.

RESPONSE

A paragraph was added P3 line 70-73.

6) In material and Methods, no need to describe again all about your 6 groups it is already in you figure 1, remove from P 1 line 71-77

RESPONSE:

The description mentioned was deleted, and the text was revised.

7) please restructure sentence on P2 L70: For instance:"… of ten specimen each, following preparation protocol presented in Figure 1

RESPONSE:

The sentence was restructured as requested.

8) In the Fig 1 please add after ER:Yag the word laser as well as in sentence P4 L109

RESPONSE

The word laser was added in Fig 1 as well as to the mentioned sentence.

9)How about ethical approval /IRB? please include in methods

RESPONSE:

Ethical approval, including the IRB number, appears in the first sentence of the Methods section. P 3 lines 89-90

10) please specify when study started and duration in the Methods

RESPONSE

This information now appears in the second sentence of the Methods section.

11) what was the reason to use "clinpro" sealant from 3M, USA as well as etching , bonding? Why not other company/country sealants?

RESPONSE:

Thank you very much for the comment.  The reason for using "clinpro" is that its results are excellent, compared to other sealants available. Importantly, the purpose of the study was to compare preparation methods rather than to compare sealing materials. Therefore, we decided to use the same materials in the various preparation groups.

12) Sample size is too small for a statistical analysis. please add paragraph on Advantages and limitations of you study

RESPONSE:

A paragraph about the limitations of the study was added, P14 L286-290 

13) P 3 line 81-84 did the team follow instruction of manufacturers of the materials used for sealant, etching bonding? If yes, please include in text

  RESPONSE:

This information was added to the Methods section, P5 L115-116

14) p 5 table 2: under E column in the total check,

  RESPONSE:

Column E was corrected on Table 2 P7

15) P 5 line 139- instead of A-E might be A-F

  RESPONSE:

This mistake was corrected, thank you.

16) p 6 line 141-142, no need of double explanation of the Fig 2, mention in the sentence only the main point.

 RESPONSE:

The sentence was removed 

17) P7 improve quality of the photos

  RESPONSE

New photos were added, we hope they will be considered satisfactory.

  18) P 8 line 200: specify which protocol?

RESPONSE: The protocol was specified, and the subject was elaborated in P 13 L 262-270.

Reviewer 2 Report

Line 9, 23: Make sure the font and size of your text are consistent throughout the manuscript.

Check line 26. Short running title: shall be in another line or after a period.

Line 40: Need improvement in English flow. "Molars and premolars are the 40teeth most susceptible to caries development, due to their occlusal surface morphology, which limits"     Materials and Methods: The materials/brand for etching, sealant, etc were not mentioned until later in a table, make sure you tell the reader that the information is below. Could using these specific materials/brands be a factor in the result? Did you try different brand materials to make sure the result was consistent and there is no bias on your technique?   ADA HEALTH FOINDATION??? check spelling   Line 93: in the following sentence, you mentioned how you did address the calibration but did not mention any result to achieve the kappa.... "To assess inter-examiner reliability (kappa between 92examiners), each examiner evaluated retention of the same 10 teeth, prior to conducting the experiment."   Line 209: No need of capital "T" here....They reported   Line 2013: Tehraniet al. check punctuation when necessary     Overall, the research is shown to be of interest as it is providing important recommendations and best practices for sealant placement. The research conducted showing the Methods in a well-organized manner, thus the results are reliable. However, as I mentioned earlier, I am concerned or wondering if the use of different materials (bonding, sealant, etching, etc) would have changed the outcomes in the results.  You need to specify the limitations of the study if you are not conducting any further experiments to be included in this manuscript.   Table 2 needs more clarification of what A,B, C, etc means as it is hard to recall from the previous paragraphs. Add the information in footnotes.   The figure 2 is blurry. improve the quality of the image.     

Author Response

Re: Manuscript ID: ijerph-886953, entitled “Laser tooth preparation for pit and fissure sealing”.

August 2020

Reviewer #2:

19)Line 9, 23: Make sure the font and size of your text are consistent throughout the manuscript.

RESPONSE:

Thank you, this was corrected

20) Check line 26. Short running title: shall be in another line or after a period.

RESPONSE:

Thank you. corrected

21) Line 40: Need improvement in English flow. "Molars and premolars are the 40teeth most susceptible to caries development, due to their occlusal surface morphology, which limits" 

RESPONSE:

This sentence was revised, P2 L57-58

22) Materials and Methods: The materials/brand for etching, sealant, etc were not mentioned until later in a table, make sure you tell the reader that the information is below.

RESPONSE:

This information is now mentioned in the text, Page 4 lines 110-114

23)  Could using these specific materials/brands be a factor in the result? Did you try different brand materials to make sure the result was consistent and there is no bias on your technique?   ADA HEALTH FOINDATION???

RESPONSE:

The focus of this research was to compare microleakage when using a number of surface tooth preparations, in the context of sealing pits and fissures. For this purpose, a conventionally and widely used resin composite sealant was chosen. We thought it important to use the same materials throughout, in order that the particular material of a certain brand company would not be responsible for the microleakage, nor the preparation itself. The authors had no conflicts of interest regarding the materials chosen.
We corrected the spelling of ADA Health Foundation.

24)check spelling   Line 93: in the following sentence, you mentioned how you did address the calibration but did not mention any result to achieve the kappa.... "To assess inter-examiner reliability (kappa between 92examiners), each examiner evaluated retention of the same 10 teeth, prior to conducting the experiment."  

 RESPONSE:

The kappa value was added.

25)Line 209: No need of capital "T" here....They reported  

RESPONSE:

This sentence was corrected.

26) Line 2013: Tehraniet al. check punctuation when necessary  

RESPONSE:

We revised this sentence.

27) Overall, the research is shown to be of interest as it is providing important recommendations and best practices for sealant placement. The research conducted showing the Methods in a well-organized manner, thus the results are reliable. However, as I mentioned earlier, I am concerned or wondering if the use of different materials (bonding, sealant, etching, etc) would have changed the outcomes in the results. 

RESPONSE:

Thank you for your comment. We addressed this issue in comment 23.

28) You need to specify the limitations of the study if you are not conducting any further experiments to be included in this manuscript.

A paragraph of limitations of the study was added in page 14 Lines 286-290

29) Table 2 needs more clarification of what A,B, C, etc means as it is hard to recall from the previous paragraphs. Add the information in footnotes. 

RESPONSE:

A footnote was added to Table 2. Page 7 line 179-185

30)The figure 2 is blurry. improve the quality of the image.   

RESPONSE:

Thank you

Reviewer 3 Report

This manuscript reports on evaluate of the retention and micro-leakage of sealants, comparing bur mechanical preparation and Er:Yag laser enamel preparation.

The topic is specific to one topic of preventive dentistry, the scope is narrow on studying one procedure and the impact is low. The quality of the work is low in that the title is about laser but the objectives and conclusions do not adhere to laser but focus on general mechanical preparation, in addition to etching. This is also exemplified by ambiguous methods description with many details missing that render the study irreproducible; hard to follow data tables and low resolution figures that lack the scientific soundness; discussion being mere listing of random literature findings that do not support the findings of the current study rather than truly discussing the Results with referring to the Tables and Figures.

In conclusion, rewrite and resubmit.

Some suggestion and detailed points to be addressed by the authors:

  1. Foremost, the title indicates the focus was evaluation of laser preparation but the manuscript did not focus on laser. Change the title.

Alternatively, keep the title: one important conclusion one can draw from the results is that with etching, laser preparation led to less micro-leakage compared with mechanical (groups B,C vs E,F) but unfortunately the authors failed to draw this conclusion.

  1. Section 3 should be incorporated into Section 2 since it is also part of Methods.

  1. “a number of other publications…” and “In their review of the literature…” interrupts the flow of general introduction. Delete them.

  1. L57 “Adding a bonding…sealant applications.” Why “the enamel that was contaminated with saliva?” This is irrelevant and appears to be borrowed directly from other publications.

  1. Table 1 “Self-cured acryl resin” inappropriate abbreviation.

Some company name does not need to use all capital letters.

  1. Detailed mechanical preparation should be given.

Which order were the experiments done? Testing of retention before thermal cycling and dying as described? The probing with a dental explorer would have loosened the sealant one would think. If no, please state it.

  1. How were The length of the sealant and the length of the dye penetration measured? How was percent of dye penetration calculated? These were not immediately available unfortunately.

  1. Table 2 is hard to follow. What are each rows of numbers?

  1. Table 3 the column of Mean+/-SE, what is the unit? 563.3+/- which value is Mean? Which value is SE?

  1. Fig. 3 is of low quality with the markings unreadable. The magnification is missing.

  1. L160 “The percentages of penetration were 100% in groups A, B and D; 39.3% in group C; 78.2% in group E; and 54.2% in group F.” This is different from the results of Fig. 2!

  1. How was cutting done exactly? How thick was each section? Was this done to ensure all three sections cut through the bottom of the pits and fissures? This is important to ensure each piece correctly represents the dye penetration hence validates the percentage penetration calculation.

  1. “However, some reports concluded that mechanical preparation was not beneficial [15,16]” was inserted randomly and the authors did not discuss this.

  1. The supplementary information does not provide additional information. No need to submit this in the future.

Author Response

Re: Manuscript ID: ijerph-886953, entitled “Laser tooth preparation for pit and fissure sealing”.

August 2020

Reviewer #3:

31) The topic is specific to one topic of preventive dentistry, the scope is narrow on studying one procedure and the impact is low. The quality of the work is low in that the title is about laser but the objectives and conclusions do not adhere to laser but focus on general mechanical preparation, in addition to etching. This is also exemplified by ambiguous methods description with many details missing that render the study irreproducible; hard to follow data tables and low resolution figures that lack the scientific soundness; discussion being mere listing of random literature findings that do not support the findings of the current study rather than truly discussing the Results with referring to the Tables and Figures. In conclusion, rewrite and resubmit.

  RESPONSE:

Thank you for your input, the article was corrected according to all the reviewers' remarks. Pits and fissure sealants are important techniques and commonly used in pediatric dentistry for preventing caries. Exploring the best method of preparation for sealant retention is an important issue.

32)  Foremost, the title indicates the focus was evaluation of laser preparation but the manuscript did not focus on laser. Change the title.

Alternatively, keep the title: one important conclusion one can draw from the results is that with etching, laser preparation led to less micro-leakage compared with mechanical (groups B,C vs E,F) but unfortunately the authors failed to draw this conclusion.

RESPONSE:

The conclusion was changed in P1 L33-36, such that it correlates better with the title.

33) Section 3 should be incorporated into Section 2 since it is also part of Methods.

RESPONSE:

Section 3 (Statistical analysis) was incorporated into section 2. 

34) “a number of other publications…” and “In their review of the literature…” interrupts the flow of general introduction. Delete them.

 RESPONSE:

These phrases were deleted. 

35) L57 “Adding a bonding…sealant applications.” Why “the enamel that was contaminated with saliva?” This is irrelevant and appears to be borrowed directly from other publications.

  RESPONSE:

Whether the addition of bonding sealants improves retention is a real clinical issue. This was the rationale in establishing groups C and F. 

 36) Table 1 “Self-cured acryl resin” inappropriate abbreviation.

RESPONSE:

The abbreviation was corrected according to the manufacturer of UNIFAST Trad.

37) Some company name does not need to use all capital letters.

RESPONSE: 

These names were corrected.

38) Detailed mechanical preparation should be given.

RESPONSE:

Details of the mechanical preparation were added on P3-4 Lines 94-99

39) Which order were the experiments done? Testing of retention before thermal cycling and dying as described?

RESPONSE:

 Yes, the order of examination was, as described: retention-thermal cycling- dye- cutting- microscope.

40) The probing with a dental explorer would have loosened the sealant one would think. If no, please state it.

RESPONSE:

 A sentence was added on P 5 lines 123-125

41) How were the length of the sealant and the length of the dye penetration measured? How was percent of dye penetration calculated? These were not immediately available unfortunately.

RESPONSE:

Clarification of the calculation was added on P6 lines 144-147.

42) Table 2 is hard to follow. What are each rows of numbers?

RESPONSE:

Footnotes were added to Table 2, to increase clarity.

 43) Table 3 the column of Mean+/-SE, what is the unit? 563.3+/- which value is Mean? Which value is SE?

RESPONSE:

The unit is µm of dye penetration, which was now added to table 3. SE=standard error. SE values were added to the table.

44)  Fig. 3 is of low quality with the markings unreadable. The magnification is missing.

RESPONSE:

The photos were replaced with photos of better quality. Magnification was added

 45) L160 “The percentages of penetration were 100% in groups A, B and D; 39.3% in group C; 78.2% in group E; and 54.2% in group F.” This is different from the results of Fig. 2!

RESPONSE:

The data referred to in the text indicate the proportion of teeth that exhibited dye penetration. Figure 2 presents the percentage of the total length of the sealant that the dye penetrated.  
We revised the wording both in the text and in the title of Figure 2, and hope it is now clearer. We also revised the relevant text in the Methods.

 46) How was cutting done exactly? How thick was each section? Was this done to ensure all three sections cut through the bottom of the pits and fissures? This is important to ensure each piece correctly represents the dye penetration hence validates the percentage penetration calculation.

RESPONSE:

This was clarified on P 6 Lines 136-138.

47) “However, some reports concluded that mechanical preparation was not beneficial [15,16]” was inserted randomly and the authors did not discuss this.

RESPONSE:

Discussion of this subject was added on Page 12 lines 228-232.

48) The supplementary information does not provide additional information. No need to submit this in the future

RESPONSE:

The supplementary information is not submitted with the revised manuscript.

Round 2

Reviewer 3 Report

After the 1st round of revision the quality of the manuscript has been improved: more methodological details have been added to increase the reproducibility; more data has been added to improve the scientific soundness and Discussion has become more relevant. The Conclusion in Abstract has been modified to correlate with the title and objectives: laser preparation.

However, the following points must be addressed.

1. The Conclusion in the main text must be modified to correlate with the title and objectives: laser preparation.

2. Table 2, title is too long. Deleted the added details of group A-F. Instead, the cell “No Penetration*” and “Penetration*” should be split in 4 with 4 row titles indicating what each of the four rows of numbers are (row-1 12, row-2 14.2, row-3 0.33 and row-4 6.67 for example). Cell “Total” should be split in 2 with 2 row titles indicating what the two rows of numbers are.

3. Table 3, with the added SE, we see that SEs are in general very big ranging 20% - 43.9% of the Mean values. Why is that? What has caused such high discrepancy?

4. Fig. 2 how come E does not have error bar?

5. Fig. 3 the same problem persists. Increase the font size of the markings so they will be visible.